# Quoted text in the mental healthcare electronic record: an analysis of the distribution and content of single-word quotations

Lasantha Jayasinghe ![ORCID],[1] Sumithra Velupillai,[1] Robert Stewart[1,2]

¹Department of Psychological Medicine, Institute of Psychiatry, Psychology and Neuroscience, King's College London, London, UK
²South London and Maudsley NHS Foundation Trust, London, UK

**Correspondence to**
Lasantha Jayasinghe;
lasantha.jayasinghe@kcl.ac.uk

## ABSTRACT

**Objective** To investigate the distribution and content of quoted text within the electronic health records (EHRs) using a previously developed natural language processing tool to generate a database of quotations.

**Design** $\chi^2$ and logistic regression were used to assess the profile of patients receiving mental healthcare for whom quotations exist. K-means clustering using pretrained word embeddings developed on general discharge summaries and psychosis specific mental health records were used to group one-word quotations into semantically similar groups and labelled by human subjective judgement.

**Setting** EHRs from a large mental healthcare provider serving a geographic catchment area of 1.3 million residents in South London.

**Participants** For analysis of distribution, 33 499 individuals receiving mental healthcare on 30 June 2019 in South London and Maudsley. For analysis of content, 1587 unique lemmatised words, appearing a minimum of 20 times on the database of quotations created on 16 January 2020.

**Results** The strongest individual indicator of quoted text is inpatient care in the preceding 12 months (OR 9.79, 95% CI 7.84 to 12.23). Next highest indicator is ethnicity with those with a black background more likely to have quoted text in comparison to white background (OR 2.20, 95% CI 2.08 to 2.33). Both are attenuated slightly in the adjusted model. Early psychosis intervention word embeddings subjectively produced categories pertaining to: mental illness, verbs, negative sentiment, people/relationships, mixed sentiment, aggression/violence and negative connotation.

**Conclusions** The findings that inpatients and those from a black ethnic background more commonly have quoted text raise important questions around where clinical attention is focused and whether this may point to any systematic bias. Our study also shows that word embeddings trained on early psychosis intervention records are useful in categorising even small subsets of the clinical records represented by one-word quotations.

## INTRODUCTION

Mental health electronic health records (EHRs) routinely capture a wealth of unstructured information detailing a patient's

## Strengths and limitations of this study

► A large sample size (n=33 499) for logistic regression of all patients receiving mental healthcare on 30 June 2019 was used in this study and comparisons made between characteristics of groups with/without a quotation.

► Pre-trained word embeddings used to categorise one-word quotations based on a large electronic mental health record corpus of around 23 million words each.

► 27% (9118) of the data for those with quotations on 30 June 2019 contained variables with missing values and therefore were not included in the adjusted logistic regression analysis.

► Investigator subjective judgement is used to determine the category label of clusters and consequently the optimum number of clusters.

clinical journey including symptoms, observed behaviours, contextual factors, assessments, interventions and outcomes within the free-text fields of case notes and correspondence.[1] The majority of studies using this information have focused on these clinical constructs.[2] However, the EHR is also a narrative account written from the perspective of healthcare professionals.[3] Within this account, clinicians often add exact quotes from patient testimony and other parties, for example, as evidence for their diagnosis or other decisions.[4 5] Quoted text in the EHR has the potential to give insight into the types of information recorded by clinicians and into the patient voice, although as secondary information filtered through the lens of the clinician, reflecting both the focus of the encounter and the individual clinician's reporting style.[6 7] This is of particular interest in two respects.

First, due to the lack of standardisation of clinical reporting,[8] it is unknown to what extent there is coherence in clinician testimony as represented by quoted text and how

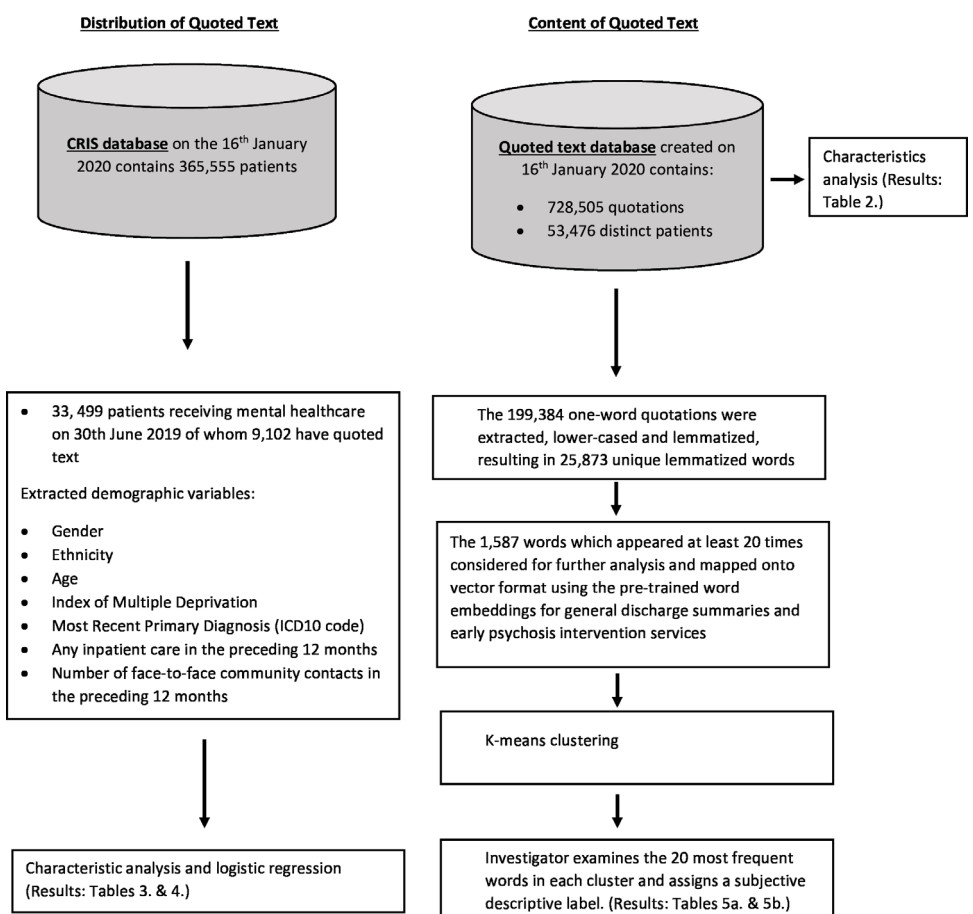

**Figure 1** Overview of methodology workflow. CRIS, clinical record interactive search. ICD10 refers to the International Classification of Diseases, 10th revision

this relates to outcomes for patients. In this respect, many previous studies, rather than examining quoted speech directly, have looked at instances of 'referencing', where the source of text is assigned to a third party using the 'he/she says' construct. In one study, there was a greater relative frequency in third-person pronoun use in a group of veterans who had died from suicide, in contrast to a comparison service-user group,[9] and another study found an increase in referencing when doctors communicated negative news to patients.[10] On the other hand, a previous study by our group found that the frequency of quoted text in the EHR did not change in the time period nearer a suicide attempt, indicating that clinicians did not change the frequency of directly reporting patient testimony even when patients symptoms may have worsened.[7]

Second, incorporating the patient voice in the EHR has become a growing area of interest[11 12] with data from new studies indicating that inclusion of electronic patient-reported outcomes (ePROs) is associated with improved levels of patient care in areas such as cancer treatment.[13] Given this context, the examination of the content of quoted text pre-existing in the EHR at least represents the beginnings of wider inclusion of the patient voice while ePROs are under development. However, little or no research has been carried out to date on such text.

As a precursor to understanding the content of quoted text, it is important to understand the patient populations from whom quotations are taken. It is currently unknown, for example, which patients are more likely to have quoted text in their EHR, or if there are variations between different demographic or diagnostic groups. Therefore, building on our previous work to ascertain quoted text at scale in the full EHR, the first objective of this study was to ascertain a fine-grain understanding of the distribution of quoted text within patients receiving secondary mental healthcare by undertaking analysis of frequencies by key demographic and clinical characteristics. Second, we sought to investigate the content of the quoted text itself. Again, to the best of our knowledge, the content of quoted text in EHRs is largely unknown, limiting conclusions that can be drawn. For example, it is unknown whether quotes predominantly relate to psycho-pathological terms that clinicians have been trained to note down, or whether they cover other indications of patient experience outside medical terminology.

Due to the large volumes of the data, we opted to approach the problem using natural language processing (NLP) methods and to apply k-means clustering to extracted text, an unsupervised method, with the aim of classifying the quotations. NLP has increasingly been

**Table 1** The word length of quotations in the database

| Word length | n | % Total |
|---|---|---|
| 0 | 1694 | 0.23 |
| 1 | 199 384 | 27.37 |
| 2 | 148 394 | 20.37 |
| 3 | 86 476 | 11.87 |
| 4 | 58 993 | 8.1 |
| 5 | 42 482 | 5.83 |
| 6 | 30 791 | 4.23 |
| 7 | 22 616 | 3.1 |
| 8 | 18 066 | 2.48 |
| 9 | 13 581 | 1.86 |
| 10–14 | 37 566 | 5.17 |
| 15–19 | 15 936 | 2.19 |
| 20–24 | 8205 | 1.13 |
| 25–29 | 5347 | 0.73 |
| 30–34 | 3854 | 0.53 |
| 35–39 | 3036 | 0.41 |
| 40–44 | 2487 | 0.35 |
| 45–49 | 2233 | 0.3 |
| 50–54 | 1951 | 0.26 |
| 55–74 | 6106 | 0.84 |
| 75–99 | 5129 | 0.7 |
| 100–149 | 6713 | 0.69 |
| 150–199 | 4028 | 0.68 |
| 200+ | 3437 | 0.68 |

**Table 2** Characteristics of patients receiving mental healthcare who have at least one quotation on the index date (16 January 2020)

| | All patients on the quotations database* | |
|---|---|---|
| | n | % Total |
| Gender | | |
| Female | 26 420 | 49.4 |
| Male | 27 035 | 50.6 |
| Unknown | 21 | 0.0 |
| Ethnicity | | |
| White European | 31 435 | 58.8 |
| Black | 14 155 | 26.5 |
| Asian | 2592 | 4.8 |
| Other | 3456 | 6.5 |
| Unknown | 1838 | 3.4 |
| Age group | | |
| 1–15 | 2463 | 4.6 |
| 16–25 | 10 363 | 19.4 |
| 26–35 | 7417 | 13.9 |
| 36–45 | 8484 | 15.9 |
| 46–55 | 9198 | 17.2 |
| 56–65 | 5942 | 11.1 |
| 66–75 | 2976 | 5.6 |
| 76–85 | 2751 | 5.1 |
| 86+ | 3872 | 7.2 |
| Unknown | 10 | 0.0 |
| Index of Multiple Deprivation | | |
| 0≤x≤20 (most deprived) | 15 386 | 28.8 |
| 20<x≤30 | 17 026 | 31.8 |
| 30<x≤93 (least deprived) | 19 323 | 36.1 |
| Unknown | 1741 | 3.3 |
| Most recent primary diagnosis via ICD10 code | | |
| F0x—organic, including symptomatic, mental disorders | 4853 | 9.1 |
| F1x—mental and behavioural disorders due to psychoactive substance use | 6065 | 11.3 |
| F2x—schizophrenia, schizotypal and delusional disorders | 8773 | 16.4 |
| F3x—mood (affective) disorders | 8773 | 16.4 |
| F4x—neurotic, stress-related and somatoform disorders | 6248 | 11.7 |
| F5x—behavioural syndromes associated with physiological disturbances and physical factors | 1958 | 3.7 |
| F6x—disorders of adult personality and behaviour | 2548 | 4.8 |
| F7x—mental retardation | 1145 | 2.1 |
| F8x—disorders of psychological developmental | 2037 | 3.8 |

used to extract clinically relevant information such as symptoms and medication from EHRs,[14] and as part of the work to investigate quoted text frequencies in relation to self-harm, we had already developed an algorithm to identify and extract these text strings from a large mental healthcare EHR.[7] One way of representing words is via word embeddings, where semantically similar words have similar numerical vector representations.[15] These vectors are generated by applying machine learning models over specific text corpora. Previous studies on the same mental healthcare platform as our data have generated word embeddings to identify terminology around general symptoms of mental illness[16] and more specifically psychosis.[17] Focusing on one-word quotations in the first instance, as the most common form, we sought to address the deficiency of information on content and to investigate the extent to which these pre-trained embeddings might be used to classify the quotations identified from a mental healthcare EHR.

## METHODS
### Study sample
The South London and Maudsley (SLaM) National Health Service (NHS) Foundation Trust (SLaM

**Table 2** Continued

| | All patients on the quotations database* | |
|---|---|---|
| | n | % Total |
| F9x—behavioural and emotional disorders with onset usually occurring in childhood and adolescence | 3578 | 6.7 |
| Zx—unspecified | 768 | 1.4 |
| Any other letter x | 376 | 0.7 |
| Not recorded | 6354 | 11.9 |
| Any inpatient care in the preceding 12 months | | |
| No | 52 986 | 99.1 |
| Yes | 490 | 0.9 |
| Number of face to face community contacts in the preceding 12 months | | |
| 0 | 40 518 | 75.8 |
| 1–7 | 5666 | 10.6 |
| 8–14 | 2521 | 4.7 |
| 15–21 | 1848 | 3.5 |
| 22–28 | 1162 | 2.2 |
| 29–42 | 1067 | 2.0 |
| 43+ | 694 | 1.3 |
| Year of first SLaM referral | | |
| 1918–2014 | 50 339 | 94.1 |
| 2015–2020 | 2580 | 4.8 |
| Unknown | 557 | 1.0 |

ICD10 refers to the International Classification of Diseases, 10th revision
*Where a patient had more than one quotation data has been derived from the date of their earliest record.
SLaM, South London and Maudsley.

provides comprehensive, near-monopoly mental healthcare services to a geographic catchment of around 1.3 m residents in four boroughs of south London, as well as some national specialist services. The mental health records used in this study were assembled using SLaM's clinical record interactive search (CRIS) platform, which currently accesses mental healthcare records for over 500 000 patients, rendering them de-identified and accessible for research use.[18]

### The distribution of individuals with quoted text
An overview of the methodology is given in figure 1. The first objective of the study was to describe the distribution of those with quoted text at the database creation date, 16 January 2020, on which there were 365 555 total active patients in SLaM from which quotations would potentially arise. The date of the first instance of quoted text for a patient was used as the index date to determine the variable values. Additionally, for comparison purposes, we extracted the same variables for all active SLaM patients on a particular index date, 30 June 2019 and compared

the people with or without quoted text to see if there were any differences.

### Variables
All variables were derived from structured text fields at index date. The basic demographic variables were gender, ethnicity and age. For analytical purposes, we summarised ethnicity into five groups, as follows: white European (British, Irish, any other white background), black (African, white and black African, Caribbean, white and black Caribbean, any other black background), Asian (Bangladeshi, Indian, Pakistani, Chinese, any other Asian background, white and Asian), other (any other mixed background, any other ethnic group) and missing. Other variables included were: the Index of Multiple Deprivation (IMD) Score, most recent primary diagnosis, whether any inpatient care was received in the preceding 12 months with a binary yes/no outcome, the number of community face to face contacts in the preceding 12 months and the year of first patient referral to SLaM.

The IMD Score[19] represents the socioeconomic status of neighbourhoods in the UK by combining various economic, social and housing indicators. It is based on the 2011 national census data and calculated from the patient's most recent address at index date and the distribution of national percentile scores for given neighbourhoods has been most commonly categorised by tertiles in previous CRIS analyses[20–22]; thus the same categories were applied here for consistency ($0 \leq x \leq 20$, $20 < x \leq 30$, $30 < x \leq 93$), with lower scored groups indicating greater deprivation. The most recent primary diagnosis was determined from the International Classification of Diseases, 10th revision (ICD10 code) assigned to the patient at index date. In this analysis, the groups were represented by the first letter and/or digit of the ICD10 code, resulting in the following categories: F0x, F1x, F2x, F3x, F4x, F5x, F6x, F7x, F8x, F9x, Zx, Any other letter x and Not recorded.

### The content of quoted text
The second objective of our study was to investigate the content of quoted text. Previous work by our team involved the development of an NLP tool to identify text occurring within quotation marks in the EHR using regular expression matching on a sample of patients hospitalised for a suicide attempt.[7] The details of the application have been previously described,[7] but in summary this algorithm yielded a precision of 0.92, recall of 0.93 against a manually annotated gold standard. As one-word quotations were the largest proportion by word count, these 199 384 instances (27% of the total) were chosen as the focus for this particular investigation.

### Statistical analysis
Data and statistical analysis were performed using standard Python (V.3.6.8) libraries. To analyse the distribution of individuals with quoted text, first the characteristics for all individuals with quoted text at the database creation date of 16 January 2020 were calculated. Where an individual

had more than one instance of quoted text, the date of the first quotation was used for variable extraction.

Second, in the sample of patients receiving mental healthcare on 30 June 2019, $\chi^2$ tests were used to test the associations between each individual variable and the presence of quoted text. $\chi^2$ tests were calculated with missing values excluded and for ordinal variables (age group, IMD, number of face to face community contacts, year of first SLaM referral) as a linear trend with one degree of freedom. Then logistic regression was used to analyse whether individual variables increased the likelihood of the presence of quotations and whether this was attenuated when all variables were incorporated in a single model. The logistic regression analyses were conducted using complete cases only.

To analyse the content of quoted text, first the one-word quotations (199 384) were extracted from the quotations database, lower-cased and lemmatised (converted to its base form) using the NLTK package, WordNetLemmatizer.[23] This resulted in 25 873 unique lemmatised words. Of these, the most frequent quotations, classified as those appearing at least 20 times, were compiled into a list for further analysis, giving 1587 words. These were then mapped into vector format via the word embeddings and used in the clustering algorithm. We used the pre-trained word embeddings generated by Viani and colleagues[17] trained specifically on CRIS records pertaining to (1) general discharge summaries (23.6 m words) reflecting all mental health disorder records (not restricted to psychosis) and (2) early psychosis intervention services (23.3 m words) across all mental health services. These were trained using a gensim[24] implementation of Word2Vec with the Continuous Bag of Words model. We felt this was a suitable approach as these embeddings have been specifically trained on CRIS records, so are more likely to reflect the semantic similarity of words in the context specific to our mental health platform. For example, words such as 'hyper' are used more loosely in general terminology, but have specific meanings in the clinical context.

The idea behind clustering methodology is to use an unsupervised algorithm to group together semantically similar words as represented by their vector forms. K-means was chosen as it has high accuracy and speed with large datasets and provides good data segmentation.[25] The implementation of k-means used was Python's *sklearn* library, which by default runs the algorithm ten times with different centroid seeds to minimise the impact of clusters forming around local minima. Several methods are available to select the optimum number of clusters. We initially sought to use the 'elbow method' to determine the number of clusters by plotting the inertia for 1–20 clusters.[26] However, as no obvious 'elbow' was apparent, we opted for silhouette analysis which examines the separation distance between each k-means cluster.[26 27] The higher the average silhouette coefficients, the further the clusters are apart, with a maximum value of 1. Under this method, the higher the average silhouette score for

each cluster, the better the representation of the data. Therefore, in statistical terms, the optimum number of clusters is represented by the highest average silhouette score over a range of possible values. This was determined by plotting the average silhouette scores for 1–20 clusters and examining the highest values. Our objective was to classify the content of one-word quotations in a way that added meaning to the group; it was therefore necessary to use subjective judgement to assess whether the optimum number of clusters provided distinct meaningful groups. To assess the usefulness of the k-means clustering algorithm in assigning semantically similar words to similar groups, the investigator examined the 20 most frequent words in each cluster and, using subjective judgement (the lead investigator's initial decision followed by consensus achieved in the research group), assigned a descriptive label for each cluster. If at least one cluster could be given a homogeneous label, then the process was complete, and this number of clusters was deemed optimum. Otherwise, the number of clusters with the next highest average silhouette score was assessed in a similar way, and so on until it was possible for the investigator to assess at least one of the clusters as a homogenous group.

## Patient involvement
We did not directly incorporate patient and public involvement (PPI) into this particular study, but the SLaM Biomedical Research Centre (BRC) Case Register used in the analysis was developed with extensive PPI and is overseen by a committee that includes service-user representatives.

## RESULTS
The previously developed tool[7] to identify quoted text from a sample of CRIS records was run over the whole CRIS database, which contained 365 555 records on 16 January 2020. After removal of blank quotes and those representing 'html' tag data, 728 505 quotations were identified from CRIS, relating to 53 476 individuals. The quotations were further categorised in terms of word count. The mean number of words in a quotation was 9, median 3, SD 25 and range 1–309, indicating a wide distribution in the size of quotations with a positive skew. Details for the volumes of quotations by word length are described in table 1.

Characteristics of patients with at least one instance of quoted text are displayed in table 2. Characteristics of the total number of patients in receipt of SLaM care on 30 June 2019 index date are further shown in table 3, alongside the subset of patients with quoted text. Quoted text was more common in male patients, in those from black compared with white ethnic groups and those living in more deprived neighbourhoods. In terms of clinical variables, quoted text was more common in those with schizophrenia, schizotypal and delusional disorders (F2x), those receiving inpatient care and those with higher numbers

**Table 3** Characteristics of patients receiving mental healthcare on the index date (30 June 2019, n=33 499) alongside a comparison of the subset of patients with at least one instance of quoted text on the 30 June 2019

| | All patients active on SLaM at 30 June 2019 | Patients with quotations active on SLaM at 30 June 2019 | | $\chi^2$ tests of independence* |
| --- | --- | --- | --- | --- |
| | n | n | % All SLaM active patients by row total | |
| **Gender** | | | | |
| Female | 15 992 | 4039 | 25.30 | $\chi^2(1)=57.81$ |
| Male | 17 459 | 5056 | 29.00 | p<0.001 |
| Missing | 48 | 7 | 14.60 | |
| **Ethnicity** | | | | |
| White European | 14 560 | 4263 | 29.30 | $\chi^2(3)=822.63$ |
| Black | 7752 | 3694 | 47.70 | p<0.001 |
| Asian | 1440 | 476 | 33.10 | |
| Other | 1864 | 488 | 26.20 | |
| Missing | 7883 | 181 | 2.30 | |
| **Age group†** | | | | |
| 1–15 | 6850 | 669 | 9.80 | $\chi^2(1)=28\ 489.0$ |
| 16–25 | 5284 | 1179 | 22.30 | p<0.001 |
| 26–35 | 5785 | 1461 | 25.30 | |
| 36–45 | 4947 | 1685 | 34.10 | |
| 46–55 | 5004 | 2108 | 42.10 | |
| 56–65 | 3083 | 1337 | 43.40 | |
| 66–75 | 1299 | 461 | 35.50 | |
| 76–85 | 882 | 166 | 18.80 | |
| 86 | 328 | 36 | 11.00 | |
| Missing | 37 | 0 | 0.00 | |
| **Index of Multiple Deprivation†** | | | | |
| 0≤x≤20 (most deprived) | 11 019 | 2205 | 20.00 | $\chi^2(1)=18\ 720.0$ |
| 20<x≤30 | 10 265 | 3066 | 29.90 | p<0.001 |
| 30<x≤93 (least deprived) | 10 963 | 3461 | 31.60 | |
| Missing | 1252 | 370 | 29.60 | |
| **Most recent primary diagnosis via ICD10 code** | | | | |
| F0x—organic, including symptomatic, mental disorders | 631 | 125 | 19.80 | $\chi^2(12)=11\ 085.74$ |
| F1x—mental and behavioural disorders due to psychoactive substance use | 2271 | 696 | 30.60 | p<0.001 |
| F2x—schizophrenia, schizotypal and delusional disorders | 5230 | 3910 | 74.80 | |
| F3x—mood (affective) disorders | 2798 | 1232 | 44.00 | |
| F4x—neurotic, stress-related and somatoform disorders | 2424 | 596 | 24.60 | |
| F5x—behavioural syndromes associated with physiological disturbances and physical factors | 543 | 144 | 26.50 | |
| F6x—disorders of adult personality and behaviour | 1165 | 611 | 52.40 | |
| F7x—mental retardation | 484 | 283 | 58.50 | |
| F8x—disorders of psychological developmental | 1418 | 413 | 29.10 | |
| F9x—behavioural and emotional disorders with onset usually occurring in childhood and adolescence | 2187 | 521 | 23.80 | |
| Zx—unspecified | 280 | 86 | 30.70 | |

Continued

**Table 3** Continued

| | All patients active on SLaM at 30 June 2019 | Patients with quotations active on SLaM at 30 June 2019 | | $\chi^2$ tests of independence* |
|---|---|---|---|---|
| | n | n | % All SLaM active patients by row total | |
| Any other letter x | 200 | 38 | 19.00 | |
| Not recorded | 13 868 | 447 | 3.20 | |
| Any inpatient care in the preceding 12 months | | | | |
| No | 33 042 | 8746 | 26.50 | $\chi^2(1)=602.52$ |
| Yes | 457 | 356 | 77.90 | p<0.001 |
| Number of face to face community contacts in the preceding 12 months† | | | | |
| 0 | 16 895 | 1589 | 9.40 | $\chi^2(1)=20\,984.0$ |
| 1–7 | 7434 | 2166 | 29.10 | p<0.001 |
| 8–14 | 3347 | 1629 | 48.70 | |
| 15–21 | 2277 | 1350 | 59.30 | |
| 22–28 | 1431 | 916 | 64.00 | |
| 29–42 | 1292 | 866 | 67.00 | |
| 43 | 823 | 586 | 71.20 | |
| Year of first SLaM referral† | | | | |
| Before 2007 | 6110 | 4494 | 73.60 | $\chi^2(1)=12\,894.0$ |
| 2007–2008 | 1932 | 1058 | 54.80 | p<0.001 |
| 2009–2010 | 1781 | 1121 | 62.90 | |
| 2011–2012 | 2028 | 989 | 48.80 | |
| 2013–2014 | 2469 | 333 | 13.50 | |
| 2015–2016 | 3789 | 331 | 8.70 | |
| 2017–2018 | 6138 | 399 | 6.50 | |
| 2019–2020 | 8995 | 178 | 2.00 | |
| Missing | 257 | 199 | 77.40 | |

ICD10 refers to the International Classification of Diseases, 10th revision
*'Missing' category has been removed when running $\chi^2$ tests.
†For ordinal variables, $\chi^2$ text for linear trends ($\chi^2(1)$) was conducted.
SLaM, South London and Maudsley.

of community face to face contacts in the preceding 12 months.

The unadjusted and adjusted logistic regression results are presented in table 4. The presence of any inpatient care in the preceding 12 months is the strongest individual indicator of quoted text, with those receiving care nearly 10 times more likely to have quoted text than those without. In terms of ethnicity, black individuals are approximately two times as likely as white Europeans to have instances of quoted text, although this is attenuated by the presence of other variables in the adjusted model. In comparison to the reference category, F2x (Schizophrenia, schizotypal and delusional disorders), other primary diagnoses are very unlikely to produce instances of quoted text. Additionally, gender, age group and IMD have very little effect on the presence of quoted text, in the adjusted model.

### Cluster analyses

The optimum number of clusters suggested by silhouette analysis (see figure 2) for the discharge summary word embeddings was 2. This yielded two clusters, which appeared to distinguish between a group referring to sentiment (negative and positive) and a miscellaneous group. As the investigator observed that both groups appeared to contain mixed rather than distinct categories, the next highest silhouette score was examined and this yielded four clusters, which are displayed in table 5. This shows that group 0 is miscellaneous with no obvious descriptive category label, while the other groups appear to contain words related to mental illness, sentiment and verbs. The optimum number of clusters using the early intervention word embeddings was 9 (see figure 3), as shown in table 6. These groups appeared to contain more clearly differentiable categories, relating to mental illness, verbs (two groups), negative sentiment, people/relationships, mixed sentiment, aggression/violence and negative connotation.

**Table 4** Unadjusted and adjusted ORs for each characteristic as an independent factor in the presence of quoted text

| | Unadjusted OR (95% CI) | Adjusted OR (95% CI)* (complete cases, n=24 381) |
|---|---|---|
| Gender (reference=Female) | | |
| Male | 1.21 (1.15 to 1.27) | 1.02 (0.94 to 1.10) |
| Ethnicity (reference=white European) | | |
| Black | 2.2 (2.08 to 2.33) | 1.28 (1.18 to 1.39) |
| Asian | 1.19 (1.06 to 1.34) | 1.03 (0.87 to 1.21) |
| Other | 0.86 (0.77 to 0.96) | 0.89 (0.77 to 1.04) |
| Age group | 1.24 (1.23 to 1.26) | 0.89 (0.87 to 0.92) |
| Index of Multiple Deprivation | 1.34 (1.30 to 1.38) | 1.01 (0.97 to 1.06) |
| Most recent primary diagnosis via ICD10 code (reference=F2x) | | |
| F0x—organic, including symptomatic, mental disorders | 0.08 (0.07 to 0.10) | 0.42 (0.32 to 0.54) |
| F1x—mental and behavioural disorders due to psychoactive substance use | 0.15 (0.13 to 0.17) | 0.33 (0.29 to 0.38) |
| F3x—mood (affective) disorders | 0.27 (0.24 to 0.29) | 0.54 (0.47 to 0.61) |
| F4x—neurotic, stress-related and somatoform disorders | 0.11 (0.10 to 0.12) | 0.38 (0.33 to 0.44) |
| F5x—behavioural syndromes associated with physiological disturbances and physical factors | 0.12 (0.10 to 0.15) | 0.51 (0.40 to 0.65) |
| F6x—disorders of adult personality and behaviour | 0.37 (0.33 to 0.42) | 0.63 (0.54 to 0.75) |
| F7x—mental retardation | 0.48 (0.39 to 0.58) | 0.57 (0.46 to 0.72) |
| F8x—disorders of psychological developmental | 0.14 (0.12 to 0.16) | 0.44 (0.37 to 0.53) |
| F9x—behavioural and emotional disorders with onset usually occurring in childhood and adolescence | 0.11 (0.09 to 0.12) | 0.4 (0.34 to 0.48) |
| Zx—unspecified | 0.15 (0.12 to 0.19) | 0.48 (0.34 to 0.67) |
| Any other letter x | 0.08 (0.06 to 0.11) | 0.27 |
| Not recorded | 0.01 (0.01 to 0.01) | 0.16 (0.14 to 0.19) |
| Any inpatient care in the preceding 12 months (reference=No) | | |
| Yes | 9.79 (7.84 to 12.23) | 3.77 (2.78 to 5.11) |
| Number of face to face community contacts in the preceding 12 months | 1.9 (1.86 to 1.93) | 1.3 (1.27 to 1.33) |
| Year of first SLaM referral | 0.52 (0.51 to 0.52) | 0.58 (0.57 to 0.59) |

ICD10 refers to the International Classification of Diseases, 10th revision
*Adjusted for gender, ethnicity, age group, index of multiple deprivation, most recent primary diagnosis, any inpatient care in the preceding 12 months, number of face to face community contacts in the preceding 12 months and year of first SLaM referral.
SLaM, South London and Maudsley.

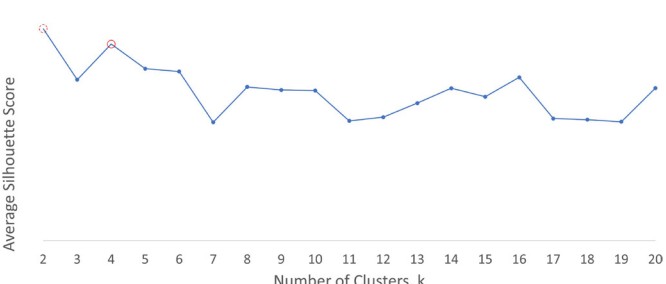

**Figure 2** Average silhouette score for clusters generated from CRIS General discharge summaries word embeddings. CRIS, clinical record interactive search.

## DISCUSSION

To the best of our knowledge, this is the first study to describe the distribution of quoted text and the content of one-word length quotations from a clinical record database, the size of which adds strength to our findings. In a sample of SLaM patients at census date 30 June 2019, those with any inpatient care in the preceding 12 months are most likely to have quoted text in the clinical record, even after adjusting for other variables. Ethnicity was the next most pertinent factor, with quoted text appearing more commonly for those with a black ethnic background, but this was attenuated slightly in the adjusted model. Individuals with schizophrenia, schizotypal and

**Table 5** The 20 most frequent words in each k-means cluster generated using the CRIS general discharge summary word embeddings and their subjective category label

| Group assigned by k-means cluster | 0 | 1 | 2 | 3 |
|---|---|---|---|---|
| Subjective label | Miscellaneous | Mental illness | Sentiment | Verbs |
| Twenty most frequent words | Friend | No | Ok | Voice |
| | Correspondence | Yes | Fine | They |
| | Right | High | Good | Nothing |
| | Boring | Breakdown | Normal | Everything |
| | Lost | Stress | Alright | People |
| | Lazy | Manic | Okay | Thing |
| | On | Problem | Happy | Help |
| | Spirit | Depression | Better | Something |
| | Bastard | Paranoia | Well | Them |
| | Hello | Mental | Safe | Control |
| | Fit | Crisis | Stable | Hate |
| | Free | Episode | Nice | It |
| | Bitch | Illness | Bad | Dead |
| | Perfect | Psychotic | Depressed | Attack |
| | Blip | Failure | Mad | Work |
| | Prn | Anxiety | Low | Kill |
| | Crap | Pressure | Paranoid | Fight |
| | Goodbye | Pain | Stuck | Know |
| | Home | Anger | Stressed | Worry |
| | Flat | Heavy | Stupid | Rule |

CRIS, clinical record interactive search.

delusional disorders (F2x) were much more likely to have quoted text than those with any other primary diagnosis, although primary diagnosis in general had little effect in the adjusted model. This study also found that one-word quotations were clustered into more distinctive categories using the early intervention word embeddings in comparison to those generated from discharge summaries. This resulted in nine groups which could be subjectively labelled as follows: mental illness, verbs (two groups), negative sentiment, people/relationships, mixed sentiment, aggression/violence and negative connotation.

As described, the relevant contexts for this study are the increasing volume of data now being routinely collected

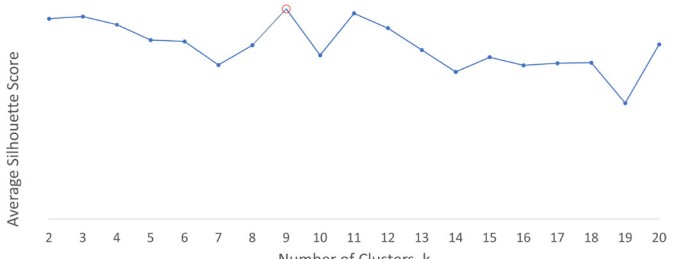

**Figure 3** Average silhouette score for clusters generated from CRIS early psychosis intervention services word embeddings. CRIS, clinical record interactive search.

in EHR resources, and the growing awareness of the potential utility for such data to support research and improved clinical practice and/or service configuration, alongside the fact that EHRs reflect primarily a clinician's perspective and authorship. Although quotations in the text remain filtered by that perspective, they do at least provide the beginnings of a 'patient voice' in the EHR while systems for direct patient input to the health record are developed. Given the lack of information on quoted text, even basic information such as the frequency of its recording and the characteristics of patients and/or services/contexts in which it is recorded, we sought to compile some preliminary data on distribution and on the content of single-word quotations as the most common type observed. This drew on earlier published work to ascertain such quotations automatically and at scale across the EHR through NLP.[7]

Our findings of any inpatient care in the preceding 12 months being the strongest indicator of quoted text may be due to those hospitalised receiving more frequent clinical observation than outpatients, leading to a greater volume of clinical notes from which quotations may arise. Further, there may be a greater focus on quoting text as evidence for decision making and medical defence practice,[4 5] given that inpatients are more likely to be suffering

**Table 6** The 20 most frequent words in each k-means cluster generated using the CRIS early psychosis intervention services word embeddings and their subjective category label

| Group assigned by k-means | 0 | 1* | 2 | 3 | 4 | 5 | 6 | 7 | 8 |
|---|---|---|---|---|---|---|---|---|---|
| Subjective label | Mental illness | Verbs | Verbs | Negative sentiment | People/ relationships | Mixed sentiment | Aggression/ violence | Healthcare | Negative connotation |
| Twenty most frequent words | No | Rule | Help | Normal | Friend | Ok | Lost | Yes | Terrible |
| | Breakdown | Break | Control | High | People | Fine | On | Correspondence | Boring |
| | Stress | Turn | Work | Depressed | Flat | Good | Rude | Mental | Horrible |
| | Problem | Go | Kill | Low | Mum | Voice | Dead | Waiting | Rubbish |
| | Depression | Cut | Know | Paranoid | Boyfriend | Alright | Gay | Should | Bastard |
| | Paranoia | Sorted | Worry | Angry | Baby | Bad | Attack | Crisis | Hello |
| | Event | Run | Stop | Hyper | Girlfriend | Okay | Fight | Carer | Fat |
| | Episode | Gone | Escape | Unwell | Family | Happy | Forced | Assessment | Free |
| | Illness | Jump | Leave | Confused | Dad | Better | Contaminated | Need | Shit |
| | Psychotic | Dropped | Treat | Manic | Mother | Mad | Bullied | Duty | Bitch |
| | Anxiety | Drop | Lie | Strange | Someone | Well | Accident | Counselling | Useless |
| | Pressure | Fall | Hurt | Stable | Old | Stuck | Missing | Therapy | Perfect |
| | Pain | Character | Feel | Odd | Partner | Stressed | Failed | Nurse | Awful |
| | Anger | Fell | Relax | Anxious | Uncle | Stupid | Attacked | Plan | Blip |
| | Heavy | | Push | Calm | Sister | They | Sectioned | Care | Prn |
| | Issue | | See | Positive | Man | Down | Hit | We | Crap |
| | Emotional | | Die | Feeling | Relationship | Sad | Lying | Urgent | Goodbye |
| | Psychosis | | Change | Suicidal | Child | Nothing | Talking | Freedom | Naughty |
| | Panic | | Do | Thought | Husband | Bored | Fighting | Respite | Flashback |
| | Seizure | | Poison | Calmer | Girl | Safe | Controlling | Trust | Rough |

*Only 14 words were present in total in this group.
CRIS, clinical record interactive search.

from the most severe mental health conditions. On a similar basis, a greater frequency of quoted text from those of black ethnicity may be explained by higher levels of psychosis being present in this group in comparison to white ethnic backgrounds[28–30] and consequently these individuals may receive a greater clinical focus.

The finding that the early psychosis intervention word embeddings produce more distinct categories in the data makes sense in the context of around 44% of all quotations in CRIS arising from patients with a recent primary diagnosis of F1x—mental and behavioural disorders due to psychoactive substance use, F2x—schizophrenia, schizotypal and delusional disorders or F3x—mood (affective) disorders, as all these have psychosis as a possible symptom. Therefore, word embeddings trained on records most similar to those from which the quotations are derived are likely to produce the best results in the clustering process. It is interesting to note that aside from categories related to mental illness and sentiment, this study has uncovered other more unexpected areas where clinicians may quote their patients, in terms of aggression/violence, people/relationships and verbs, indicating an emphasis on the circumstances of a patient's experience rather than purely symptomatology.

### Strengths and limitations

This study has several strengths. First it examines a novel area by focussing on the distribution and content of quotations within the EHR rather than the full record, giving insight into information clinicians may quote beyond clinical terminology. Furthermore, the large sample sizes for analysis of both distribution and content add significance to the findings. Additionally, the word embeddings used to represent the one-word quotations have been trained on millions of words which are highly relevant since they have been derived specifically from mental health records on the same platform.

The findings of our study need to be taken with several limitations in mind. One limitation of our study is that categories applied may be heterogeneous, for example, the ethnic groups selected. Another limitation is that 27% of the sample data were incomplete cases and therefore were not included in the adjusted logistic regression analysis. Another consideration is that data for logistic regression were examined at one point in time, so unknown confounders may be present in the data, such as previous service use for a different mental health disorder. In terms of investigating the content of quoted text, one key limitation is that the labelling of groups found by clustering is subjective and based on the assessment of the researcher. Another key limitation is that what is found in the text is dependent on what the clinician notes down; this will be subject to training and individual preferences and biases. Additionally, attribution of the speaker is not determined by the algorithm although the majority of quotations were from patients.[7] Further, as we chose to investigate one-word quotations as a first step, the meaning derived from words in terms of clustering is limited without context.

Therefore, further studies should examine longer strings of quotations to gauge a better understanding of content. Additionally, further studies could use contextual word-vector representations. Under this methodology, words are assigned vector representations based on the surrounding contextual words, to give a better idea of how a specific word is used in a particular context.

## CONCLUSIONS

Despite limitations, this is an important study as the first of its kind to investigate the profile of patients and the areas of patient experience that are highlighted in quoted speech within the clinical record. The successful creation of a database across all CRIS to identify quoted speech is a first step in making this data available for future studies. The findings that inpatients and those from a black ethnic background more commonly have quoted text raise important questions around where clinical attention is focused and whether this may point to any systematic bias. Our study also shows that word embeddings trained on early psychosis intervention records are useful in categorising small subsets of the clinical records represented by one-word quotations.

**Contributors** RS and LJ conceived the study design with advice from SV. LJ wrote the paper and analysed the data. RS provided supervisory guidance. All authors provided critical input for the paper and approved the submission. RS is the author acting as guarantor for the study.

**Funding** LJ, SV and RS are part-funded by the National Institute for Health Research (NIHR) Biomedical Research Centre at the South London and Maudsley NHS Foundation Trust and King's College London. RS is additionally funded by a Medical Research Council (MRC) Mental Health Data Pathfinder Award to King's College London; an NIHR Senior Investigator Award; the National Institute for Health Research (NIHR) Applied Research Collaboration South London (NIHR ARC South London) at King's College Hospital NHS Foundation Trust. The views expressed are those of the authors and not necessarily those of the NIHR or the Department of Health and Social Care.

**Competing interests** RS has received research support in the last 5 years from Janssen, GSK and Takeda.

**Patient and public involvement** Patients and/or the public were not involved in the design, or conduct, or reporting, or dissemination plans of this research.

**Patient consent for publication** Not applicable.

**Ethics approval** CRIS, as a data resource for secondary analysis, has IRB approval from the Oxford Research Ethics Committee C (reference 18/SC/0372).

**Provenance and peer review** Not commissioned; externally peer reviewed.

**Data availability statement** Data are available upon reasonable request. Data must remain within the SLaM firewall and any requests to access the data can be addressed to cris.administrator@kcl.ac.uk.

**ORCID iD**
Lasantha Jayasinghe http://orcid.org/0000-0003-3907-2645

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
