## [Reviewer comments · BMJ Open]

ARTICLE DETAILS

TITLE (PROVISIONAL)	Quoted Text in the Mental Health Care Electronic Record: An Analysis of the Distribution and Content of Single-Word Quotations
AUTHORS	Jayasinghe, Lasantha; Velupillai, Sumithra; Stewart, Robert

VERSION 1 – REVIEW

REVIEWER	Silva, Tatiana Universidade Federal de Pernambuco, neuropsiquiatria
REVIEW RETURNED	21-Mar-2021

GENERAL COMMENTS	Although the topic in question is relevant and current, the problematization, contextualization, methodological path and results are not presented in a clear way. Below are listed questions about the topics mentioned that need to be resolved in the text: Problematization - despite pointing out the need for research, the authors do not clearly mention the real problem that will be dealt with in the text and cite an issue that is too broad and that was not addressed in detail in its methodology and results; Contextualization: although it is clear that the topic at hand (mental health) is highly relevant, the context has not been explored in a sufficiently clear way to justify the need and relevance of the study to the scientific community Methodological path: because it is a retrospective analysis and based on citations content, the authors can create a flowchart or figure that helps the understanding of all phases from data extraction to statistical analysis Results - the data seem to have been presented in a crude way (for example the variable age - it was not regrouped so that the table would be shorter and more objective) it is suggested to present the data in a more objective and direct way.
--

REVIEWER	Tayefi, Maryam Norwegian Centre for E-health Research
REVIEW RETURNED	22-Mar-2021

GENERAL COMMENTS	 1. you have used the turtles for IMD, do you have any reference for that? please refer. you can try the quartiles (Q1, median and Q3) and check which category gives better results. 2. Please provide the reference for "elbow method" in cluster analysis 3. what is the silhouette analysis? please provide the reference. 4. you have mentioned in table 3 there is comparison between patients with and without the quoted text, but in the table we will
---

	see the comparison between all patients and the patients at least one quoted text. please clarify that. 5. In table 5a and 5b, how do you select the subjective labels?please explain more.
--	---

VERSION 1 – AUTHOR RESPONSE

Reviewer: 1

Although the topic in question is relevant and current, the problematization, contextualization, methodological path and results are not presented in a clear way. Below are listed questions about the topics mentioned that need to be resolved in the text:

Problematization - despite pointing out the need for research, the authors do not clearly mention the real problem that will be dealt with in the text and cite an issue that is too broad and that was not addressed in detail in its methodology and results.

Response: We are sorry that this was unclear. We have added substantial text to the Introduction which we hope now frames more clearly and explicitly the problems being addressed, namely the lack of research to date on the frequency and distribution of quoted text in the EHR, and the lack of research to date on the content of such quotations.

Contextualization: although it is clear that the topic at hand (mental health) is highly relevant, the context has not been explored in a sufficiently clear way to justify the need and relevance of the study to the scientific community

Response: Again, we are sorry that this was unclear in the original version and hope that the need and relevance of the study are now more explicitly delineated in the revised and extended Introduction section, as well as in the Discussion. We opted for BMJ Open for this submission because of the editorial policy around publishing research which is ethical and valid (which we believe applies to our study) rather than filtering on the perceived impact of the findings. We are aware that quoted text in EHRs is a relatively discrete topic to be investigating; however, we reiterate that it has received little or no research to date (to our knowledge) and thus our study was developed as a first step in this field.

Methodological path: because it is a retrospective analysis and based on citations content, the authors can create a flowchart or figure that helps the understanding of all phases from data extraction to statistical analysis

Response: We have added and incorporated a flowchart to give an overview of the methodology workflow and we hope that this is what the Reviewer had in mind.

Results: the data seem to have been presented in a crude way (for example the variable age - it was not regrouped so that the table would be shorter and more objective) it is suggested to present the data in a more objective and direct way.

Response: As this is a large sample and a descriptive paper on which we anticipate further research might build, we felt that it was most appropriate to include a fine-grain description. We apologise for the lack of clarity and have added a paragraph to the Introduction to explain this element of the rationale further.

Reviewer: 2

1. You have used the turtles for IMD, do you have any reference for that? Please refer. You can try the quartiles (Q1, median and Q3) and check which category gives better results.

Response: We have used tertiles for the IMD score as this has been the standard approach employed in categorising IMD scores in this data resource and we would be keen to retain the approach for consistency with other reports unless there was a compelling a priori reason to adopt an alternative classification. Text and new reference citations (to some of the other studies using the same approach) have been added to the Methods > Variables paragraph.

2. Please provide the reference for "elbow method" in cluster analysis.

Response: This reference has been added to the manuscript. (Methods > Statistical analysis, paragraph 4)

3. What is the silhouette analysis? Please provide the reference.

Response: Further text has been added to clarify this approach, including a further two references. (Methods > Statistical analysis, paragraph 4)

4. You have mentioned in table 3 there is comparison between patients with and without the quoted text, but in the table we will see the comparison between all patients and the patients at least one quoted text. Please clarify that.

Response: We apologise that the title to the Table 3 was misleading, and we have changed it to avoid confusion as well as adding a more detailed description of the table in the Results section, paragraph 2. We also hope that the new flow diagram clarifies the approach.

5. In table 5a and 5b, how do you select the subjective labels? Please explain more.

Response: The explanation for how these labels are determined is given in Methods -> Statistical analysis, last paragraph, and further text has been added to confirm that this is based on a subjective judgement with a consensus approach.